# Benzo[1,2-*d*:4,5-*d*′]bis([1,2,3]thiadiazole) and Its Bromo Derivatives: Molecular Structure and Reactivity

**DOI:** 10.3390/ijms24108835

**Published:** 2023-05-16

**Authors:** Timofey N. Chmovzh, Daria A. Alekhina, Timofey A. Kudryashev, Rinat R. Aysin, Alexander A. Korlyukov, Oleg A. Rakitin

**Affiliations:** 1N. D. Zelinsky Institute of Organic Chemistry, Russian Academy of Sciences, 119991 Moscow, Russiatp12345678@yandex.ru (T.A.K.); 2Nanotechnology Education and Research Center, South Ural State University, 454080 Chelyabinsk, Russia; 3Higher Chemical College, Mendeleev University of Chemical Technology of Russia, 125047 Moscow, Russia; 4Department of Chemistry, Moscow State University, 119899 Moscow, Russia; 5A. N. Nesmeyanov Institute of Organoelement Compounds, Russian Academy of Sciences, 119334 Moscow, Russia; aysin.rinat@gmail.com (R.R.A.); alex@xrlab.ineos.ac.ru (A.A.K.)

**Keywords:** sulfur-nitrogen heterocycles, 4-bromobenzo[1,2-*d*:4,5-*d*′]bis([1,2,3]thiadiazole), X-ray analysis, ab initio calculations, EDDB and GIMIC methods, UV-Vis spectra, aromatic nucleophilic substitution, Suzuki and Stille cross-coupling reactions, direct (het)arylation palladium-catalyzed reactions

## Abstract

Benzo[1,2-*d*:4,5-*d*′]bis([1,2,3]thiadiazole) (isoBBT) is a new electron-withdrawing building block that can be used to obtain potentially interesting compounds for the synthesis of OLEDs and organic solar cells components. The electronic structure and delocalization in benzo[1,2-*d*:4,5-*d*′]bis([1,2,3]thiadiazole), 4-bromobenzo[1,2-*d*:4,5-*d*′]bis([1,2,3]thiadiazole), and 4,8-dibromobenzo[1,2-*d*:4,5-*d*′]bis([1,2,3]thiadiazole) were studied using X-ray diffraction analysis and ab initio calculations by EDDB and GIMIC methods and were compared to the corresponding properties of benzo[1,2-*c*:4,5-*c*′]bis[1,2,5]thiadiazole (BBT). Calculations at a high level of theory showed that the electron affinity, which determines electron deficiency, of isoBBT was significantly smaller than that of BBT (1.09 vs. 1.90 eV). Incorporation of bromine atoms improves the electrical deficiency of bromobenzo-bis-thiadiazoles nearly without affecting aromaticity, which increases the reactivity of these compounds in aromatic nucleophilic substitution reactions and, on the other hand, does not reduce the ability to undergo cross-coupling reactions. 4-Bromobenzo[1,2-*d*:4,5-*d*′]bis([1,2,3]thiadiazole) is an attractive object for the synthesis of monosubstituted isoBBT compounds. The goal to find conditions for the selective substitution of hydrogen or bromine atoms at position 4 in order to obtain compounds containing a (het)aryl group in this position and to use the remaining unsubstituted hydrogen or bromine atoms to obtain unsymmetrically substituted isoBBT derivatives, potentially interesting compounds for organic photovoltaic components, was not set before. Nucleophilic aromatic and cross-coupling reactions, along with palladium-catalyzed C-H direct arylation reactions for 4-bromobenzo[1,2-*d*:4,5-*d*′]bis([1,2,3]thiadiazole), were studied and selective conditions for the synthesis of monoarylated derivatives were found. The observed features of the structure and reactivity of isoBBT derivatives may be useful for building organic semiconductor-based devices.

## 1. Introduction

The aromaticity and reactivity of heterocyclic compounds are among the most studied problems in organic chemistry [1,2,3,4,5]. The properties of monocyclic heterocycles have been thoroughly studied for a long time, while fused heterocyclic systems often face a number of problems with their aromaticity/antiaromaticity and the consequent differences in reactivity [1,2]. Fused heterocyclic systems containing many nitrogen and chalcogen (mainly sulfur) atoms, which have pronounced acceptor properties, in the rings attracted particular interest in recent years [6]. Electron-accepting moieties are widely represented in π-conjugated organic molecules in various combinations with electron donors and π-conjugated bridges. These organic chromophores are widely used in semiconductor-based devices such as dye-sensitized solar cells (DSSCs), organic field-effect transistors (OFETs), organic light-emitting diodes (OLEDs), and electrochromic devices (ECDs) [7]. Hybridization of the energy levels between the donor and acceptor parts in molecules can decrease the difference between E_HOMO_ and E_LUMO_ (the energy band gap, E_gap_), thus improving the optoelectronic properties of the molecule [8]. The important feature of the acceptor fragment is the electron affinity (*EA*) that is related to the energies of the lowest unoccupied molecular orbital (*E*_LUMO_).

A variety of heterocyclic acceptors are well known and intensely studied [9,10,11]. Heterocycles with a high electron affinity as electron-acceptor blocks have received a lot of attention [12]. An exceptional place among these heterocycles is occupied by 2,1,3-benzothiadiazole (**BTD**) and their derivatives (Figure 1) due to their excellent properties, such as strong electron-withdrawing properties, intense light absorption, and good photochemical stability [13,14]. Nevertheless, there is a strong demand to create electron-deficient heterocyclic systems with a stronger accepting ability. One of the options is to annulate the **BTD** cycle with another thiadiazole ring in order to build benzo[1,2-*c*:4,5-*c*′]bis[1,2,5]thiadiazole (benzo-bis-thiadiazole, **BBT**), a sulfur–nitrogen heterocycle with the lowest LUMO energy [15]. In fact, **BBT** derivatives are being actively explored for application in pharmacology and in various optoelectronic devices [16]. On the other hand, replacement of the 1,2,5-thiadiazole ring in 2,1,3-benzothiadiazole (**BTD**) by the 1,2,3-thiadiazole ring results in benzo[*d*][1,2,3]thiadiazole (**isoBTD**) compounds with properties similar to those of **BTD** but with a higher E_LUMO_ and band gap (Eg) [17] values. It was recently found that a **BBT** isomer, benzo[1,2-*d*:4,5-*d*′]bis([1,2,3]thiadiazole) (**isoBTD**), also had promising electron-accepting properties, and its 4,8-dibromo derivative could be successfully involved in aromatic nucleophilic substitution reactions and in Suzuki–Miyaura and Stille palladium-catalyzed cross-coupling reactions with selective formation of mono- and bis-arylated heterocycles [18]. Many sulfur-containing heterocycles are known to possess anticancer activity [19] due to low-lying C-S σ* or C-N σ* orbitals, which are responsible for drug–target interactions [20]. Therefore, **isoBBT** derivatives are of additional interest in terms of potential biological activity.

In order to reveal the electronic structure of **isoBBT** derivatives and find a possibility to obtain mono-arylated **isoBBT** derivatives selectively, we describe a study of the electronic structure and electron delocalization in benzo[1,2-*d*:4,5-*d*′]bis([1,2,3]thiadiazole), its 4-bromo and 4,8-dibromo derivatives by X-ray analysis and ab initio calculations using the electron density of delocalized bonds (EDDB) [21,22] and the gauge-including magnetically induced currents (GIMIC) [23,24] methods, as well as palladium-catalyzed C-H direct arylation reactions, and C-Br aromatic nucleophilic and cross-coupling reactions for 4-bromobenzo[1,2-*d*:4,5-*d*′]bis([1,2,3]thiadiazole) with selective formation of mono-arylated derivatives.

## 2. Results and Discussion

### 2.1. Structure and Electronic Percularities of Benzo[1,2-d:4,5-d′]bis([1,2,3]thiadiazole) and Its Bromo Derivatives

According to X-ray diffraction studies, the conformations of **1**–**3** in the crystal are planar (Figure 2), and the lengths of all chemical bonds fall in the range typical of thiadiazole derivatives. The molecules of **1** and **3** occupy the positions at the crystallographic center of inversion, so the unique parts of their unit cells are two halves of the corresponding molecules. The small number or absence of hydrogen atoms in **1**–**3** allow one to assume that the contribution of hydrogen bonds to the respective lattice energies are low. Indeed, all N...H and S...H distances in **1** and **2** exceed the sums of the Van der Waals radii of these elements. Thus, other types of intermolecular interactions are responsible for the stabilization of the crystal structures of **1**–**3**. The planar molecules of **1**–**3** are assembled into infinite stacks. Each crystallographically independent molecule in **1** and **3** forms separate stacks (Figure 3 and Figure 4). The stacks are interlinked by weak N...S and N...Br interactions.

The distribution of electrostatic potential mapped on the molecular Hirshfeld surfaces [25,26] of **1**–**3** indicates that the N...S and N...Br interactions are complementary and maintain the crystal packing stability (Figure 5). Two-dimensional fingerprint plots (Appendix A) for **1**–**3** indicate that they made a relatively large contribution to the intermolecular contacts with d_e_ = d_i_ ≈ 2 Å, which is characteristic for N…S, N…Br and stacking interactions. In turn, the nature of stacking interactions can be described as the interaction of the N=N bond with the central benzene rings (positive and negative regions of electrostatic potential).

To compare the electron-deficiency of compounds **1**–**3** and **BBT**, the *EA* values were calculated at the MP2 and bt-STEOM-CCSD levels (for details, see Section 3.3*. Calculations Details*) (Table 1, Appendix A). Although *EA* values calculated at the bt-STEOM-CCSD level should be more accurate, the *EA* MP2 values for **1**–**3** differ by no more than 0.1 eV. The difference between the adiabatic and vertical *EA* values is not large (up to ~0.15 eV), which indicates a weak structural rearrangement upon electron attachment. The adiabatic *EA* value of **isoBBT** (**1**) is 1.09 eV, which is significantly lower than that of **BBT** (1.90 eV). Thus, the desymmetrization of **BBT** leads to a strong decrease in its electron deficiency. In series **1**–**3**, the Br substituent exhibits electron acceptor properties and significantly increases the *EA* values of each Br substituent by ~0.2 eV. The HOMO–LUMO gap (*E_gap_*) variation in series **1**–**3** and **BBT** is opposite to the *EA* variation, which is not surprising for aromatic polycyclic molecules.

The UV-Vis spectra of **1**–**3** (Figure 6) show the longest wavelength band with an ill-defined vibrational resolution in the region of 375–425 nm, which corresponds to the π-π* HOMO–LUMO transition (the orbital shapes are given in Appendix A in ESI). This band, which corresponds to *E_gap_*, gradually red-shifts from **1** to **2** and **3** and increases in intensity, indicating a conjugation of the Br substituent with the π-orbital system [27].

Two modern criteria, EDDB [21,22] and GIMIC [23,24], at the MP2 level of theory have been applied to estimate the conjugation and aromaticity in molecules **1**–**3** (Table 2, Appendix A). The π-conjugation system of **BBT** and **isoBBT** consists of 14 ē on 7 orbitals (see Appendix A in ESI), which obey the Hückel rule. The **BBT** and **isoBBT** molecules are π-aromatic, as confirmed by EDDB and GIMIC results. The IC distributions in **1**–**3** (Figure 7) are typical of aromatic rings, but the IC density is distributed unevenly. The current strength (IRCS value) for the six-membered ring (~15 nA/T) is greater than that for the five-membered ring (~11 nA/T) despite the different ring size and the presence of three types of ring currents: local (for each ring), semilocal (over two adjacent rings), and global (over three rings) [28]. The difference in IRCS values indicates a significantly greater degree of delocalization in the six-membered formally benzene ring compared to the heterocyclic five-membered ring. This also leads to the manifestation of a local diamagnetic current of the six-membered ring, which is well distinguishable in the ring annulation region (Figure 7), which is usually non-observable.

Numerical analysis of the degree of delocalization by the EDDB method revealed that the total number of effective delocalized π-electrons (π-EDDB**_H_** value in Table 2) increased slightly from **1** to **2** and **3**, which agrees with the UV-vis spectra. The low sensitivity of the EDDB method compared to optical spectra is worthy of note [28]. Analysis of the local conjugation effects by comparing the EDDB values for the C_6_ and C_6_Br fragments (the π-EDDB**_F_** values in Table 2) showed that the participation of a lone pair of the Br atoms in the conjugation was 0.39–0.48 ē, which corresponds to ~8% of the delocalization degree in the central ring. A comparison of the π-EDDB**_F_** values for the five- and six-membered rings indicates that the delocalization in the thiadiazole ring is smaller by 6–8%, which qualitatively agrees with the GIMIC data. Upon Br substitution, the small decrease of aromaticity degree in the six-membered ring (π-EDDB**_F_** in the C_6_ ring changes by 0.06 ē) is compensated by a small aromaticity increase in thiadiazole rings (π-EDDB**_F_** in the C_2_N_2_S ring changes by 0.1 ē). Thus, the overall delocalization increase from **1** to **2** and **3** is caused by the participation of lone pairs of Br in conjugation, while the change in π-aromaticity is small.

It is worthy of note that the global aromaticity degree of **1**–**3** is significantly weaker than that of **BBT,** as evidenced by both π-EDDB**_H_** and IRCS values (Table 2). Moreover, the delocalization in **BBT** is stronger in five-membered rings in contrast to **isoBBT**. The same trend is observed for **isoBTD** and **BTD** (see Appendix A in ESI). This difference in electron delocalization is caused by the lower degree of aromaticity of the 1,2,3-thiadazole ring compared to that of the 1,2,5-thiadazole ring being fused with the benzene ring. Apparently, the significantly lower *EA* value for **isoBBT** as compared to **BBT** is due to the difference in delocalization.

### 2.2. Aromatic Nucleophilic Substitution of 4-Bromobenzo[1,2-d:4,5-d′]bis([1,2,3]thiadiazole) ***2***

The main goal of this stage of the work was to develop the optimal conditions for obtaining the products of nucleophilic aromatic substitution reactions (S_N_Ar) in 4-bromobenzo[1,2-*d*:4,5-*d*′]bis([1,2,3]thiadiazole) **2** using various aromatic and aliphatic *O*-, *S*-, and *N*-nucleophiles and to compare these conditions with the reactions of dibromo derivative **3** [18].

We have studied the reactions of replacement of the bromine atom in the benzene ring of **2** for amino groups in order to obtain substitution products. It was shown that the reaction of 4-bromobenzo[1,2-*d*:4,5-*d*′]bis([1,2,3]thiadiazole) **2** with two equivalents of morpholine in DCM at room temperature for 12 h gave monoamine derivative **4a** in trace amounts (Table 3, entry 1). To increase the yield of compound **4a**, we studied various conditions for this chemical reaction. It was found that the nature of the solvent significantly affected the yield of the final product by changing the rate of the reaction. Using TLC analysis, we showed that morpholine nearly did not react with monobromo derivative **2** in MeCN at room temperature within 12 h (Table 3, entry 2), compared to DMF (Table 3, entry 3), and gave mono-substitution product **4a** in 6 and 15% yields, respectively. Refluxing the reaction mixture in MeCN for 24 h with two equivalents of morpholine gave mono-substitution product **4a** in 85% yield. Through heating in DMF at 80 °C, a complete conversion of the initial dibromide was observed within 12 h with the formation of mono-substitution product **4a** in 65% yield (Table 3, entry 5). Thus, the optimal conditions for the synthesis of unsymmetrical compound **4a** that was involved the treatment of bromo derivative **2** with two equivalents of morpholine in refluxing MeCN (Table 3, entry 4) was extended to other primary and secondary amines. It was found that piperidine **5b** and pyrrolidine **5c** reacted with bromide **2** to form substitution products **4** in high yields (Table 3, entries 6–7). It should be noted that attempts to perform the reaction with cyclopentaindole **5d** failed due to the decomposition of monobromide **2** into a mixture of unidentifiable compounds (Table 3, entry 8). Bromide **2** reacted with primary amines, for example, with aniline **5e**, on heating at 130 °C in DMF to form substitution product **4e** in moderate yield (Table 3, entry 9). However, with aliphatic primary amines, such as cyclohexylamine **5f** and *tert*-butylamine **5g**, the reaction resulted in partial decomposition of the starting bromide **2**, even on heating to 80 °C in DMF (Table 3, entries 10–13).

Thus, we have shown that nucleophilic substitution reactions of monobromide **2** occurred more slowly than those of dibromide **3**. For example, complete conversion of dibromide **3** by treatment with morpholine occurred within 18 h [18], while monobromide **2** reacted in 24 h; the same trend was observed for piperidine and pyrrolidine. However, in the case of aniline, it was necessary to heat the reaction mixture in DMF to high temperatures. Monobromide **2** reacted only when the reaction mixture was heated to 130 °C for 24 h to give a monosubstitution product in 40% yield, while dibromide **3** reacted at 100 °C in 18 h to give a mono-substitution product with a higher yield of 50%. It should be noted that upon incorporation of an amine into the molecule of dibromide **3**, the rate of the substitution reaction of the second bromine atom sharply decreased, which required the use of more drastic conditions, namely, prolonged heating in DMF at 130 °C [18]. Based on this, we can conclude that the reactivity of monobromide **2** is between those of dibromide **3** and mono-substituted amino derivatives. In addition, monobromide **2** did not react with cyclopentaindoline **5d** since complete decomposition of the starting bromide **2** into a mixture of unidentifiable compounds was observed, while dibromide **3** reacted with it to give only the mono-substitution product [18].

A study of the reactions of monobromide **2** with such *S*-nucleophiles as thiophenol, hexynethiol, and dodecanethiol showed that they occurred similarly to the reactions of these nucleophiles with dibromide **3** [18] in the presence of sodium hydride in tetrahydrofuran at room temperature to give monomercapto derivatives **6** in high yields (Figure 1).

4-Bromobenzo[1,2-*d*:4,5-*d*′]bis([1,2,3]thiadiazole) **2**, like 4,8-dibromo derivative **3**, was found to be resistant to various *O*-nucleophiles*,* such as water, methanol, ethanol, phenol, and the corresponding sodium alcoholates. We have shown that monobromide **2** did not react with water upon heating in either THF or DMSO at 80 °C. It was shown that in all cases 4-bromobenzo[1,2-*d*:4,5-*d*′]bis([1,2,3]thiadiazole) **2** did not react with them, either in THF or in DMF. Heating of the reaction mixtures did not result in the nucleophilic substitution of bromine atoms but, rather, resulted only in partial decomposition of the starting compound **2**.

### 2.3. Cross-Coupling Reactions of 4-Bromobenzo[1,2-d:4,5-d′]bis([1,2,3]thiadiazole) ***2***

We studied the Suzuki–Miyaura reaction of 4-bromobenzo[1,2-*d*:4,5-*d*′]bis([1,2,3]thiadiazole) **2** with various aromatic and heteroaromatic pinacolate esters of boronic acids **7**. The selection of conditions for the Suzuki reaction was based on the example of the reaction with thiophene pinacolate ester **7a**. We have shown that the nature of the reagents, solvents, and the temperature of the reaction medium significantly affect the course of the reactions. The tetrakis(triphenylphosphine)–palladium complex (Pd(PPh_3_)_4_) was used, as it is the most widely used catalyst in these reactions, and potassium carbonate was used as the base. It was shown that when the reaction was carried out at 110 °C in toluene for 24 h, mono-coupling product **8a** was isolated in 60% yield. (Table 4, entry 1). At the same time, the addition of water increased the yield of product **8a** to 70%, which is apparently due to the solubility of the base (K_2_CO_3_) in water (Table 4, entry 2). Replacement of toluene with dioxane or xylene did not increase the yield of the cross-coupling product **8a** (Table 4, entries 3–4). Thus, the highest yield of the mono-coupling product was achieved in the toluene–water medium. These conditions were extended to other organoboron esters **7**; the yields of mono-coupling products **8** (Table 4, entries 5–11) varied from 64% to 70%.

Thus, it was shown that the Suzuki reaction of 4-bromobenzo[1,2*d*:4,5*d*′]bis([1,2,3]thiadiazole) **2** and boronic esters **7** gave the best results with a toluene/water mixture due to the fact that monobromo derivative **2** is much more hydrolytically stable than dibromo derivative **3**, while water promotes the cross-coupling reaction by dissolving the inorganic base. A comparison of the Suzuki cross-coupling times of dibromide **3** [18] and monobromide **2** showed that the reactions for dibromide **3** occurred better and slightly faster under anhydrous conditions than in the presence of small amounts of water. In contrast, owing to the high hydrolytic stability of monobromide **2**, the Suzuki reactions involving the latter occurred under aqueous conditions rather than under anhydrous conditions. In addition, to replace both bromine atoms in the molecule of dibromide **3**, more drastic conditions were required, i.e., heating the reaction mixture in xylene at 130 °C, which, in turn, was also due to a decrease in the reactivity of monobromo derivatives upon incorporation of thienyl or phenyl substituents.

The Stille reaction of 4-bromobenzo[1,2-*d*:4,5-*d*′]bis([1,2,3]thiadiazole) **2** was studied with various aromatic and heteroaromatic stannyl derivatives **9a**–**h**. The optimal conditions for the reactions were found using the example of a reaction with thienyltributyl stannane **9a** in the presence of PdCl_2_(PPh_3_)_2_, a catalyst that is widely used in these reactions. The reaction with reflux in toluene in the presence of 1.2 equivalents of stannane **9a** gave monoaryl derivative **8a** in 75% yield (Table 5, entry 1). On replacement of toluene with THF or dioxane, a decrease in the yield of the target product **8a** was observed (Table 5, entries 2,3). The best conditions were applied to other aryl (hetaryl) stannanes **9**. As a result, we obtained a number of mono-coupling products **8a**–**h** in good yields (Table 5, entries 4–10).

It was previously found that in the Stille reaction of dibromide **3** and stannyl derivatives **9** under mild conditions (heating in toluene at 60 °C), only one bromine atom was replaced, whereas to replace the bromine atom in the molecule of monobromide **2**, the reaction mixture had to be heated in toluene at 110 °C. Similar conditions were also required for incorporation of both thienyl and phenyl substituents into the molecule of dibromide **3**. It should be noted that the yields of the Stille reaction products were similar for both monobromide **2** and dibromide **3** and varied from 50% to 73%.

### 2.4. Palladium-Catalyzed C-H (Het)arilation Reactions of 4-Bromobenzo[1,2-d:4,5-d′]bis([1,2,3]thiadiazole) ***2***

Direct C–H arylation reactions are a modern, environmentally attractive method for building a C–C bond with two aromatic and/or heteroaromatic compounds, which allows the number of steps in this process to be reduced, avoiding the use of toxic (e.g., organotin) and flammable (e.g., butyl lithium) derivatives that are used, for example, in the Stille or Suzuki reactions [29,30,31]. For **BTD** heterocyclic systems, three methods are known for the synthesis of 4,7-disubstituted **BTD**: (1) the reaction of 4,7-dibromobenzo[*c*][1,2,5]thiadiazoles with arenes and heteroarenes; (2) the reaction of 4,7-unsubstituted benzo[*c*][1,2,5]thiadiazoles with halogeno (bromo- or iodo-) arenes and heteroarenes; and (3) oxidative direct arylation of 4,7-unsubstituted benzo[*c*][1,2,5]thiadiazoles with arenes and heteroarenes. All of the above reactions are catalyzed by palladium compounds. The application of the direct C–H arylation method for 4-bromobenzo[1,2-*d*:4,5-*d*′]bis([1,2,3]thiadiazole) **2** could selectively produce monoaryl derivatives and further a wide range of unsymmetrical 4,7-disubstituted **isoBBT** derivatives (especially push–pull compounds), which are in great demand as components of various optoelectronic devices [9,32]. The synthesis of monoaryl derivatives from dibromo**BTD** using the Suzuki and Stille reactions is often difficult due to the formation of hard-to-separate mixtures of the starting compound with mono- and bis-arylation products [33]. Only a few examples of direct C–H het-arylation reactions have been described for 4-bromobenzo[*c*][1,2,5]thiadiazole **10** by method (1) with thiophene [34] and furan derivatives (Figure 2) [35].

Only one example of direct C–H hetarylation of tricyclic benzo-bis-thiadiazoles is described in the literature: the reaction of benzo[1,2-*d*:4,5-*d*′]bis([1,2,3]thiadiazole) **1** with 2-bromothiophene **11** in the presence of palladium(II) acetate, potassium pivalate, and di-*tert*-butyl(methyl) salt of phosphonium tetrafluoroborate (PBu^t^_2_Me HBF_4_) in toluene at 120 °C to give 4,8-bis(5-(triisopropylsilyl)thiophen-2-yl)benzo[1,2-*d*:4,5-*d*′]bis([1,2,3]thiadiazole) **12** in a low yield (Figure 3) [36].

Considering the small number of published articles in the field of direct C–H arylation of heterocyclic systems based on **BTD**, we decided to use all the three methods for the **isoBBT** systems, namely, to study the replacement of bromine and hydrogen atoms in monobromo **BBT 2** and the oxidative C–H arylation of **2** with the purpose of synthesizing mono-substituted **BBT** derivatives as starting compounds to obtain asymmetric diaryl derivatives of the **BBT** heterocyclic system.

#### 2.4.1. Palladium-Catalyzed C–H Activation Reactions of 4-Bromobenzo[1,2-*d*:4,5-*d*′]bis([1,2,3]thiadiazole) **2** with Haloaromatic and Heteroaromatic Compounds (Replacement of a Hydrogen Atom in **2**)

We studied the feasibility of the addition of aromatic and heteroaromatic halogen derivatives to 4-bromobenzo[1,2-*d*:4,5-*d*′]bis([1,2,3]thiadiazole) **2** under the conditions of the C–H activation reaction to selectively obtain mono-coupling products **14**. The development of the optimal reaction conditions was performed using the reaction with 2-bromo-5-(2-ethylhexyl)thiophene **13d** in the presence of various palladium catalysts and organic ligands as an example. It was shown that the nature of the palladium catalyst, ligand, solvent, and temperature of the reaction significantly affected the results of the reactions (Table 6). Refluxing in toluene in the presence of palladium acetate (Pd(OAc)_2_) and potassium pivalate (KOPiv) resulted in partial decomposition of the starting tricycle **2** without formation of the target product **14d** (Table 6, entry 1). Incorporation of ligands, such as tri-*tert*-butylphosphine (Bu^t^_3_P) or bis(diphenylphosphino)ferrocene (dppf), also did not trigger the cross-coupling reaction (Table 6, entries 3,4). In contrast, incorporation of XPhos led to the formation of a mono-coupling product **14d** in 7% yield (Table 6, entry 2). The addition of catalysts such as tetrakis(triphenylphosphine)palladium (Pd(PPh_3_)_4_), tris(dibenzylideneacetone)dipalladium (Pd_2_(dba)_3_), and bis(triphenylphosphine)palladium chloride (PdCl_2_(PPh_3_)_2_) also did not favor the reactions (Table 6, entries 5, 6, and 8). The use of a catalytic system based on (Pd(OAc)_2_) and PBu^t^_2_Me·HBF_4_ made it possible to increase the yield of target C–H activation product **14d** considerably. In fact, the reaction in refluxing toluene gave mono-coupling product **14d** in a good yield of 65% (Table 6, entry 10). Replacement of toluene with xylene at 140 °C did not increase the yield of compound **14d** (Table 6, entry 11). We have shown that the C–H activation reaction did not affect the carbon atom bound to the bromine atom, which, in turn, makes it possible to obtain monobromo derivatives in moderate yields. The optimal conditions for the cross-coupling reaction developed by us were extended to aromatic and heteroaromatic derivatives **14b**–**h**. While the C–H activation reactions with bromothiophene compounds **13a**–**d** occurred selectively and in moderate yields (Table 6, entries 13–16), the reactions with aryl bromides occurred with much greater difficulty. The replacement of aryl bromides with aryl iodides increased the yields of mono-coupling products **14e h** significantly: the use of the Pd(OAc)_2_ and PBu^t^_2_Me·HBF_4_ catalytic system in refluxing toluene gave the target products **14** in moderate yields (Table 6, entries 13, 18–22).

#### 2.4.2. Palladium-Catalyzed C-H Activation Reactions of 4-Bromobenzo[1,2-*d*:4,5-*d*′]bis([1,2,3]thiadiazole) **2** with Aromatic and Heteroaromatic Compounds (Replacement of Bromine Atom in **2**)

We began to study the reaction of tricycle **2** with thiophene using the conditions developed for 4-bromobenzo[*c*][1,2,5]thiadiazole **10** (see Figure 2). First of all, we found that Pd_2_dba_3_ did not catalyze the reaction of 4-bromobenzo[1,2-*d*:4,5-*d*′]bis([1,2,3]thiadiazole) **2** with (2-ethylhexyl)thiophene **15d** in the presence of bases, such as cesium and potassium pivalates, and various phosphine ligands in toluene; the starting heterocycle was isolated from the reaction mixtures in high yields. Therefore, the main attention was further paid to the catalysis of this reaction with palladium acetate (Table 7). It was shown that the use of a catalytic system of palladium acetate (Pd(OAc)_2_) and potassium pivalate in the reaction of tricycle **2** and **15d** gave product **8d** in a moderate yield (Table 7, entry 1). The incorporation of ligands such as tri-*tert*-butylphosphine (Bu^t^_3_P), bis(diphenylphosphino)ferrocene (dppf), and XPhos did not trigger the cross-coupling reaction (Table 7, entries 2–4). The use of a catalytic system based on Pd(OAc)_2_ and PBu^t^_2_Me·HBF_4_ in toluene also did not result in the formation of target product **8d** (Table 7, entry 5). Nevertheless, an increase in the temperature of the reaction mixture to 130 °C led to the formation of product **8d** in a moderate yield (Table 7, entry 6). Unexpectedly, the reaction performed in DMA (see ref. [36]) resulted only in the decomposition of the starting dibromide **2** (Table 7, entry 7). The reaction conditions that we developed were extended to other thiophene derivatives **8a**–**c** (Table 7, entries 9–11). Attempts to carry out the C–H arylation reaction with aromatic compounds, such as toluene or xylene using various catalytic systems, failed: the starting tricycle **2** was isolated in high yields.

#### 2.4.3. Palladium-Catalyzed Oxidative C-H Thienylation Reactions of 4-Bromobenzo[1,2-*d*:4,5-*d*′]bis([1,2,3]thiadiazole) **2** with Thiophenes

We studied the reactions of oxidative C–H (het)arylation of monobromide **2** with (2-ethylhexyl)thiophene **15d** using silver oxide (Ag_2_O) as the oxidizing agent in anhydrous DMSO under the conditions reported for 2,1,3-benzothiadiazole [37,38]. It was shown that when palladium trifluoroacetate was used as the catalyst, the C–H activation reaction did not occur (Table 8, entry 1); after stirring for 24 h the starting compound **2** was isolated in high yield. Replacing palladium trifluoroacetate with palladium acetate at 110 °C resulted in successful activation of the reaction that produced bromoaryl derivative **14d** in 68% yield (Table 8, entry 2). The use of such silver salts, such as silver acetate (AgOAc), silver nitrate (AgNO_3_), silver tetrafluoroborate (AgBF_4_), silver perchlorate (AgClO_4_), and Ag_2_O in anhydrous DMSO gave no results (Table 8, entries 4–6). An increase in the temperature of the reaction mixture to 120 °C did not increase the yield of target product **14d** (Table 8, entry 3). The conditions we found were extended to other thiophene derivatives **15a,b,c** to obtain the corresponding thienylated products **14** in low to moderate yields (Table 8, entries 7–9).

Thus, depending on the chosen catalytic system, it is possible to perform the selective syntheses of mono-derivatives **8** and **14**, both with a C–H component and with a bromine atom for further transformations.

## 3. Materials and Methods

### 3.1. Materials and Reagents

The chemicals were purchased from the commercial sources (Sigma-Aldrich, St. Louis, MO, USA) and used as received. 4-Bromobenzo[1,2-*d*:4,5-*d*′]bis([1,2,3]thiadiazole) **2** [39], thiophene pinacolate esters **7a**–**d** [40], tributyl-2-thienylstannanes **9a**–**d** [41], and tributylarylstannanes **9e**–**h** [42] were prepared according to the published methods and characterized by NMR spectra. All synthetic operations were performed under a dry argon atmosphere. The solvents were purified by distillation over the appropriate drying agents.

### 3.2. Analytical Instruments

The melting points were determined on a Kofler hot-stage apparatus and were uncorrected. ^1^H and ^13^C NMR spectra were recorded on a Bruker AM-300 instrument (Bruker Ltd., Moscow, Russia) with TMS as the standard. *J* values are given in Hz. MS spectra (EI, 70 eV) were obtained with a Finnigan MAT INCOS 50 instrument (Thermo Finnigan LLC, San Jose, CA, USA). High-resolution MS spectra were measured on a Bruker micrOTOF II instrument using electrospray ionization (ESI). IR spectra were measured with a Bruker “Alpha-T” instrument (Bruker, Billerica, MA, USA) in KBr pellets. UV-vis spectra in the region 200–900 nm were registered for DMF solutions of **1**–**3** C= 10^–4^ M in the standard 10 mm quartz cell using a Carl Zeiss Specord M400 spectrophotometer.

### 3.3. Calculations Details

Geometry optimization, calculation of NAO and DMNAO matrices for EDDB analysis [21,22], calculation of magnetic shielding matrices for GIMIC analysis [23,24] at MP2(fc) theory level with the cc-pVTZ basis set were performed in the Gaussian program [43]. Additionally, an unrestricted MP2-level geometry optimization of anion radicals was performed to calculate adiabatic electron affinity (EA) values. The vertical and adiabatic EA values were also calculated at the bt-STEOM-CCSD level using TightPNO and RIJCOSX approximations using the ORCA [44] program for molecules with optimized geometry at the MP2(fc) level. The ZPVE correction for EA values is calculated at the TPSS-D4/cc-pVTZ level using the ORCA program. The positive EA values given in Table 1 correspond to the profitability of electron attachment to **1**–**5**. The RunEDDB [21] script was used for EDDB analysis, the π-EDDB_H_ value corresponds to a whole molecule, and π-EDDB**_F_** is of a selected fragment. The calculation of ring currents (IC) by the GIMIC method was performed using the GIMIC 2.0 program [23,45]. The ring current strength (IRCS) values were calculated by integrating with respect to the N–S and C–C bonds for 5- and 6-membered cycles by Appendix A in ESI, according to the recommendations [45]. The orbitals were visualized using the Avogadro program [46], the π-EDDB isosurfaces and IC distribution maps were constructed using the ParaView program [47]. The IC located below 0.5 Å on the molecular plane (Figure 2) are omitted for clarity of diatropic ICs from the π-cloud contribution.

### 3.4. X-ray Crystallography

X-ray diffraction data for **1** were collected with Bruker Quest diffractometer while the data for **2** and **3** were collected at 100 K on a four-circle Rigaku Synergy S diffractometer using graphite Mo K_α_-radiation. The intensity data were integrated and corrected for absorption and decay by APEX 3 (Bruker QUEST) and CrysAlisPro software (Austin, TX, USA, accessed on 1 September 2022) [48]. The structures were solved by dual-space algorithm using SHELXT and refined on *F^2^* using SHELXL-2018 [49] in anisotropic approximation [50] for non-hydrogen atoms. All hydrogen atoms were placed in ideal calculated positions and refined as riding atoms with relative isotropic displacement parameters. A rotating group model was applied for methyl groups. The structure **1** was refined as two component non-merohedral twin with TWINABS program implemented in APEX3 software. The scale factors for twin components are equal to 0.6673(15) and 0.3327(15). The twinning for **3** was established with Olex2 software, the scale factors for components are 0.521(3) and 0.479(3). The Cambridge Crystallographic Data Centre contains all crystallographic data for this paper (deposition numbers: 2255420, 2256390, and 2210625). These data can be obtained free of charge via http://www.ccdc.cam.ac.uk/conts/retrieving.html (accessed on 10 April 2023) (or from the CCDC, 12 Union Road, Cambridge CB2 1EZ, UK; Fax: +44-1223-336033; E-mail: deposit@ccdc.cam.ac.uk). Detailed information related to the X-ray diffraction studies of **1**–**3** is summarized in Appendix A.

### 3.5. Synthesis of Compounds

#### 3.5.1. General Procedure for the Preparation of Aminated Products **4a**–**c**

Amine **5a**–**c** (0.36 mmol) was added to a solution of 4-bromobenzo[1,2-*d*:4,5-*d*′]bis([1,2,3]thiadiazole) **2** (50 mg, 0.18 mmol) in dry MeCN (10 mL) at room temperature, and the mixture was stirred at reflux for 24 h, poured into water (20 mL), and extracted with CH_2_Cl_2_ (3 × 35 mL). The combined organic layers were washed with brine, dried over MgSO_4_, filtered, and concentrated under reduced pressure. The crude product was purified by column chromatography (silica gel Merck 60).

4-(benzo[1,2-*d*:4,5-*d*′]bis([1,2,3]thiadiazole)-4-yl)morpholine (**4a**)

Orange solid, 39 mg (85%), eluent–CH_2_Cl_2_/hexane, 1:2 (*v*/*v*). R_f_ = 0.2 (CH_2_Cl_2_/hexane, 1:1, (*ν*/*ν*)). Mp = 195–197 °C. IR *ν*_max_ (KBr, cm^–1^): 2961, 2918, 1851, 1563, 1537, 1441, 1388, 1339, 1290, 1262, 1243, 1108, 1067, 1025, 980, 925, 848,835, 809, 680, 611, 514. ^1^H NMR (300 MHz, CDCl_3_): *δ* 8.72 (s, 1H), 4.06–3.96 (m, 8H). ^13^C NMR (100 MHz, CDCl_3_): *δ* 160.2, 150.3, 142.3, 139.6, 129.2, 104.4, 67.4, 52.2. HRMS (ESI-TOF), *m*/*z*: calcd for C_10_H_9_N_5_OS_2_ [M]^+^, 279.0243; found, 279.0240. MS (EI, 70 eV), *m*/*z* (*I*, %): 279 ([M]^+^, 6), 251 (20), 165 (55), 93 (4), 69 (80), 28 (100).

4-(Piperidin-1-yl)benzo[1,2-*d*:4,5-*d*′]bis([1,2,3]thiadiazole) (**4b**)

Orange solid, 39 mg (80%), eluent–CH_2_Cl_2_/hexane, 1:1 (*v*/*v*). R_f_ = 0.5 (CH_2_Cl_2_/hexane, 1:1, (*ν*/*ν*)). Mp = 155–157 °C. IR *ν*_max_ (KBr, cm^–1^): 2928, 2847, 2809, 1565, 1534, 1443, 1387, 1341, 1288, 1243, 1222, 1123, 1086, 975, 844, 810, 678, 609, 625, 609, 543. ^1^H NMR (300 MHz, CDCl_3_): *δ* 8.59 (s, 1H), 3.98–3.88 (m, 4H), 1.96–1.98 (m, 6H). ^13^C NMR (100 MHz, CDCl_3_): *δ* 160.2, 149.8, 142.4, 140.7, 129.0, 102.6, 53.4, 26.8, 24.4. HRMS (ESI-TOF), *m*/*z*: calcd for C_11_H_11_N_5_S_2_ [M]^+^, 277.0450; found, 277.0444. MS (EI, 70 eV), *m*/*z* (*I*, %): 277 ([M]^+^, 45), 249 (90), 220 (6), 192 (46), 165 (30), 96 (40), 69 (100), 41 (95), 27 (80).

4-(Pyrrolidin-1-yl)benzo[1,2-*d*:4,5-*d*′]bis([1,2,3]thiadiazole) (**4c**)

Orange solid, 37 mg (79%), eluent–CH_2_Cl_2_/hexane, 1:2 (*v*/*v*). R_f_ = 0.4 (CH_2_Cl_2_/Hexane, 1:1, (ν/ν)). Mp = 196 –198 °C. IR *ν*_max_ (KBr, cm^–1^): 2969, 2950, 2875, 2856, 1635, 1555, 1538, 1450, 1388, 1337, 1315, 1278, 1194, 1037, 982, 844, 807, 669, 640, 528. ^1^H NMR (300 MHz, CDCl_3_): *δ* 8.19 (s, 1H), 4.39–4.28 (m, 4H), 2.23–2.14 (m, 4H). ^13^C NMR (100 MHz, CDCl_3_): *δ* 160.5, 145.4, 143.3, 138.8, 124.8, 97.5, 53.6, 26.1. HRMS (ESI-TOF), *m*/*z*: calcd for C_10_H_9_N_5_S_2_ [M]^+^, 263.0294; found, 263.0295. MS (EI, 70 eV), *m*/*z* (*I*, %): 263 ([M]^+^, 8), 235 (15), 206 (5), 192 (3), 178 (6), 96 (13), 69 (100), 41 (50), 27 (55), 18 (10).

N-Phenylbenzo[1,2-*d*:4,5-*d*′]bis([1,2,3]thiadiazole)-4-amine (**4e**) 

Aniline **5e** (33 mg, 0.36 mmol) was added to a solution of 4-bromobenzo[1,2-*d*:4,5-*d*′]bis([1,2,3]thiadiazole) **2** (50 mg, 0.18 mmol) in dry DMF (10 mL), and the mixture was stirred at 130 °C for 24 h, poured into water, and extracted with CH_2_Cl_2_ (3 × 35 mL). The combined organic layers were washed with water, brine, dried over MgSO_4_, filtered, and concentrated under reduced pressure. The crude product was purified by column chromatography (silica gel Merck 60). Red solid, yield 20 mg (40%), eluent–CH_2_Cl_2_/hexane, 1:1 (*v*/*v*)). R_f_ = 0.4 (CH_2_Cl_2_/hexane, 1:1 (*v*/*v*)). Mp = 157–160 °C. IR *ν*_max_ (KBr, cm^–1^): 3344, 1581, 1546, 1495, 1449, 1429, 1385, 1343, 1298, 1255, 1072, 854, 816, 764, 702, 677, 538. ^1^H NMR (300 MHz, CDCl_3_): *δ* 8.59 (s, 1H), 8.33 (br. s, 1H), 7.56–7.45 (m, 3H), 7.31 (d, *J* = 7.1, 2H). ^13^C NMR (100 MHz, CDCl_3_): *δ* 161.3, 146.7, 141.3, 137.2, 134.7, 129.8, 127.8, 126.9, 121.4, 101.7. HRMS (ESI-TOF), *m*/*z*: calcd for C_12_H_8_N_5_S_2_ [M + H]^+^, 286.0216; found, 286.0221. MS (EI, 70 eV), *m*/*z* (*I*, %): 285 ([M]^+^, 4), 257 (55), 228 (4), 196 (5), 185 (30), 160 (12), 153 (13), 125 (10), 93 (14), 77 (60), 69 (100), 51 (58).

#### 3.5.2. General Procedure for the Reaction of 4-Bromobenzo[1,2-*d*:4,5-*d*′]bis([1,2,3]thiadiazole) **2** with Thiols

Sodium hydride (7.2 mg, 0.18 mmol) was added to a solution of thiol (0.18 mmol) in dry THF (15 mL) at 0 °C with stirring. The reaction mixture was stirred at 0 °C for 30 min, then 4-bromobenzo[1,2-*d*:4,5-*d*′]bis([1,2,3]thiadiazole) **2** (50 mg, 0.18 mmol) was added. The mixture was stirred for 6 h at room temperature. On completion (monitored by TLC), the mixture was poured into water (20 mL) and extracted with CH_2_Cl_2_ (3 × 5 mL). The combined organic layers were washed with brine, dried over MgSO_4_, filtered, and concentrated under reduced pressure. The crude product was purified by column chromatography.

4-(Phenylthio)benzo[1,2-*d*:4,5-*d*′]bis([1,2,3]thiadiazole) (**6a**)

Yellow solid, 43 mg (80%), Mp = 140–142 °C, eluent–CH_2_Cl_2_/hexane, 1:2 (*v*/*v*). R_f_ = 0.5 (CH_2_Cl_2_/hexane, 1:1 (*v*/*v*)). IR *ν*_max_ (KBr, cm^–1^): 1638, 1578, 1475, 1438, 1388, 1330, 1286, 1196, 1075, 1019, 901, 866, 832, 800, 734, 686, 660, 541, 515, 470. ^1^H NMR (300 MHz, CDCl_3_): *δ* 9.11 (s, 1H), 7.65 (d, *J* = 7.4, 2H), 7.60–7.54 (m, 1H), 7.46 (t, *J* = 7.4, 2H). ^13^C NMR (75 MHz, CDCl_3_): *δ* 158.6, 155.5, 140.2, 139.0, 135.7, 130.6, 129.9, 129.2, 127.4, 111.3. HRMS (ESI-TOF), *m*/*z*: calcd for C_12_H_6_N_4_S_3_Ag [M + Ag]^+^, 408.8800; found, 408.8790. MS (EI, 70 eV), *m*/*z* (*I*, %): 304 ([M + 2]^+^, 2), 303 ([M + 1]^+^, 3), 302 ([M + 1]^+^, 35), 274 (70), 245 (8), 214 (14), 201 (100), 177 (90), 170 (65), 158 (85), 145 (50), 133 (53), 124 (80), 109 (90), 100 (92), 93(95), 76 (94), 70 (95), 65 (94), 52 (94), 45 (96), 39 (95), 27 (55).

4-(Hexylthio)benzo[1,2-*d*:4,5-*d*′]bis([1,2,3]thiadiazole) (**6b**)

Yellow solid, 41 mg (75%), eluent–CH_2_Cl_2_/hexane, 1:4 (*v*/*v*). R_f_ = 0.7 (CH_2_Cl_2_/hexane, 1:1 (*v*/*v*)). Mp = 112–115 °C. IR *ν*_max_ (KBr, cm^–1^): 2956, 2924, 2853, 1615, 1510, 1456, 1391, 1334, 1285, 1196, 1104, 901, 868, 798, 752, 721, 658, 543. ^1^H NMR (300 MHz, CDCl_3_): *δ* 9.12 (s, 1H), 3.68 (t, *J* = 7.3, 2H), 1.67 (p, *J* = 7.3, 2H), 1.49–1.40 (m, 2H), 1.27–1.21 (m, 4H), 0.84 (t, *J* = 6.7, 3H). ^13^C NMR (75 MHz, CDCl_3_): *δ* 157.1, 156.4, 140.8, 126.5, 125.8, 111.5, 36.8, 31.2, 30.0, 28.2, 22.4, 13.9. HRMS (ESI-TOF), *m*/*z*: calcd for C_12_H_14_N_4_S_3_Ag [M + Ag]^+^, 416.9426; found, 416.9425. MS (EI, 70 eV), *m*/*z* (*I*, %): 311 ([M + 1]^+^, 10), 310 ([M]^+^, 66), 311 ([M − 1]^+^, 15), 281 (60), 225 (55), 197 (35), 183 (65), 125 (36), 93 (58), 69 (100), 41 (80), 29 (79).

4-(Dodecylthio)benzo[1,2-*d*:4,5-*d*′]bis([1,2,3]thiadiazole) (**6c**)

Green solid, 53 mg (76%), eluent–CH_2_Cl_2_/hexane, 1:2 (*v*/*v*). R_f_ = 0.8 (CH_2_Cl_2_/hexane, 1:1 (*v*/*v*). Mp = 31–33 °C. IR *ν*_max_ (KBr, cm^–1^): 2954, 2920, 2849, 1653, 1469, 1390, 1287, 1195, 903, 871, 833, 798, 718, 656, 542. ^1^H NMR (300 MHz, CDCl_3_): *δ* 9.11 (s, 1H), 3.67 (t, *J* = 7.3, 2H), 1.67 (p, *J* = 7.3, 2H), 1.48–1.36 (m, 2H), 1.29–1.16 (m, 16H), 0.86 (t, *J* = 6.6, 3H). ^13^C NMR (75 MHz, CDCl_3_): *δ* 157.1, 156.4, 143.0, 140.8, 125.8, 111.4, 36.8, 31.9, 30.0, 29.6, 29.59, 29.55, 29.5, 29.4, 29.0, 28.5, 22.7, 14.1. HRMS (ESI-TOF), *m*/*z*: calcd for C_18_H_26_N_4_S_3_Na [M + Na]^+^, 417.1212; found, 417.1212. MS (EI, 70 eV), *m*/*z* (*I*, %): 396 ([M + 2]^+^, 5), 395 ([M + 1]^+^, 10), 394 ([M]^+^, 46), 355 (100), 341 (70), 295 (44), 281 (90), 221 (85), 207 (49), 69 (50), 55 (40).

#### 3.5.3. General Procedure for the Preparation of Arylated Benzo-bis-thiadiazoles **8** under Suzuki Coupling Conditions (Procedure A)

A mixture of 4-bromobenzo[1,2-*d*:4,5-*d*′]bis([1,2,3]thiadiazole) **2** (50 mg, 0.18 mmol), boronic ether **16a**–**h** (0.18 mmol), K_2_CO_3_ (24 mg, 0.18 mmol), 1 mL (H_2_O), and Pd(PPh_3_)_4_ (20 mg, 10% mmol) in dry toluene (8 mL) was degassed by argon and heated at 110 °C in a sealed vial. On completion (monitored by TLC), the mixture was poured into water and extracted with CH_2_Cl_2_ (3 × 35 mL). The combined organic layers were washed with brine, dried over MgSO_4_, filtered, and concentrated in vacuo. The crude product was purified by column chromatography.

#### 3.5.4. General Procedure for the Preparation of Arylated Benzo-bis-thiadiazoles **8** under Stille Coupling Conditions (Procedure B)

PdCl_2_(PPh_3_)_2_ (12 mg, 10% mmol) and stannane **20a**–**h** (0.21 mmol) were added to a solution of 4-bromobenzo[1,2-*d*:4,5-*d*′]bis([1,2,3]thiadiazole) **2** (50 mg, 0.18 mmol) in anhydrous toluene (4 mL) The resulting cloudy, yellow mixture was stirred and degassed by argon in a sealed vial. The resulting yellow mixture was then stirred at 60 °C for the desired time. On completion (monitored by TLC), the mixture was washed with water and the organic layer was extracted with CH_2_Cl_2_ (3 × 35 mL), dried over MgSO_4_, and then concentrated in vacuo. The crude product was purified by column chromatography.

#### 3.5.5. General Procedure for the Synthesis of Arylated Benzo-bis-thiadiazoles **8** by Palladium-catalyzed C-H Direct Arylation Reaction of 4-Bromobenzo[1,2-*d*:4,5-*d*′]bis([1,2,3]thiadiazole) **2** (Procedure C)

A mixture of 4-bromobenzo[1,2-*d*:4,5-*d*′]bis([1,2,3]thiadiazole) **2** (50 mg, 0.18 mmol), thiophene **3a**–**e** (0.20 mmol), Pd(OAc)_2_ (6 mg, 15% mmol), pivalic acid (20 mg, 0.20 mmol), and K_2_CO_3_ (27 mg, 0.20 mmol) were added to an air-free flask, which was then purged in dry toluene (8 mL), degassed by argon, and heated at 110 °C in a sealed vial. On completion (monitored by TLC), the mixture was poured into water and extracted with CH_2_Cl_2_ (3 × 35 mL). The combined organic layers were washed with brine, dried over MgSO_4_, filtered, and concentrated under reduced pressure. The crude product was purified by column chromatography.

4-(Thiophen-2-yl)benzo[1,2-*d*:4,5-*d*′]bis([1,2,3]thiadiazole) (**8a**)

Orange solid, 34 mg (70%, Procedure A), 34 mg (70%, Procedure B), or 18 mg (37%, Procedure C), eluent–CH_2_Cl_2_/hexane, 1:2 (*v*/*v*). R_f_ = 0.3 (CH_2_Cl_2_/hexane, 1:1, (*ν*/*ν*)). Mp = 173–150 °C. IR ν_max_ (KBr, cm^–1^): 1636, 1532, 1437, 1432, 1393, 1328, 1286, 1258, 1142, 858, 812, 715, 666, 544. ^1^H NMR (300 MHz, CDCl_3_): *δ* 9.19 (s, 1H), 8.25 (d, *J* = 3.7, 1H), 7.75 (d, *J* = 5.0, 1H), 7.43–7.32 (m, 1H). ^13^C NMR (100 MHz, CDCl_3_): *δ* 158.4, 154.0, 140.9, 138.9, 137.6, 131.6, 130.5, 129.8, 128.6, 112.1. HRMS (ESI-TOF), *m*/*z*: calcd for C_10_H_5_N_4_S_3_ [M + H]^+^, 276.9671; found, 276.9663. MS (EI, 70 eV), *m*/*z* (*I*, %): 276 ([M]^+^, 6), 248 (75), 220 (10), 176 (11), 151 (100), 93 (25), 69 (95), 45 (12), 28 (5).

4-(4-Hexylthiophen-2-yl)benzo[1,2-*d*:4,5-*d*′]bis([1,2,3]thiadiazole) (**8b**)

Yellow solid, 44 mg (69%, Procedure A), 46 mg (71%, Procedure B), or 23 mg (37%, Procedure C), eluent–CH_2_Cl_2_/hexane, 1:2 (*v*/*v*). R_f_ = 0.4 (CH_2_Cl_2_/hexane, 1:1, (*ν*/*ν*)). Mp = 65–68 °C. IR ν_max_ (KBr, cm^–1^): 2956, 2924, 2853, 1640, 1540, 1513, 1494, 1451, 1398, 13754, 1333, 1287, 1249, 1188, 1081, 967, 854, 815, 775, 725, 661, 615, 522. ^1^H NMR (300 MHz, CDCl_3_): *δ* 9.16 (s, 1H), 8.14 (s, 1H), 7.34 (s, 1H), 3.18–2.61 (m, 2H), 1.79–1.70 (m, 2H), 1.40–1.30 (m, 6H), 0.91 (t, *J* = 8.0, 3H). ^13^C NMR (100 MHz, CDCl_3_): *δ* 157.2, 152.8, 144.1, 138.9, 136.2, 136.1, 132.2, 124.3, 122.5, 110.6, 30.7, 29.6, 29.5, 28.0, 21.6, 13.1 HRMS (ESI-TOF), *m*/*z*: calcd for C_16_H_17_N_4_S_3_ [M + H]^+^, 361.0610; found, 361.0606. MS (EI, 70 eV), *m*/*z* (*I*, %): 362 ([M + 2]^+^, 3), 361 ([M + 1]^+^, 6), 360 ([M]^+^, 50), 332 (100), 248 (20), 235 (19), 220 (12), 165 (18), 120 (13), 69 (60), 43 (57), 29 (48).

4-(5′-(Trimethylsilyl)-[2,2′-bithiophen]-5-yl)benzo[1,2-*d*:4,5-*d*′]bis([1,2,3]thiadiazole) (**8c**)

Red solid, 54 mg (70%, Procedure A), 52 mg (68%, Procedure B), or 31 mg (40%, Procedure C), eluent–CH_2_Cl_2_/hexane, 1:2 (*v*/*v*). R_f_ = 0.3 (CH_2_Cl_2_/hexane, 1:1, (*ν*/*ν*)). Mp = 155–157 °C. IR ν_max_ (KBr, cm^–1^): 2958, 2924, 2853, 1724, 1641, 1494, 1464, 1364, 1279, 1263, 1187, 1081, 968, 892, 818, 725, 486. ^1^H NMR (300 MHz, CDCl_3_): *δ* 9.15 (s, 1H), 8.15 (d, *J* = 4.0, 1H), 7.44 (d, *J* = 3.5, 1H), 7.40 (d, *J* = 4.0, 1H), 7.22 (d, *J* = 3.5, 1H), 0.37 (s, 9H). ^13^C NMR (100 MHz, CDCl_3_): *δ* 158.6, 153.6, 143.1, 142.2, 141.2, 141.05, 136.8, 135.9, 135.2, 132.6, 126.5, 124.9, 123.1, 111.7, 0.00(TMS). HRMS (ESI-TOF), *m*/*z*: calcd for C_17_H_15_N_4_S_4_Si [M + H]^+^, 430.9943; found, 430.9928. MS (EI, 70 eV), *m*/*z* (*I*, %): 432 ([M + 2]^+^, 1), 431 ([M + 1]^+^, 2), 430 ([M]^+^, 8), 402 (7), 305 (6), 200 (10), 175 (12), 93 (45), 69 (100), 45 (30).

4-(5-(2-Ethylhexyl)thiophen-2-yl)benzo[1,2-*d*:4,5-*d*′]bis([1,2,3]thiadiazole) (**8d**)

Orange solid, 47 mg (68%, Procedure A), 48 mg (70%, Procedure B), or 38 mg (55%, Procedure C), eluent–CH_2_Cl_2_/hexane, 1:2 (*v*/*v*). R_f_ = 0.4 (CH_2_Cl_2_/hexane, 1:1, (*ν*/*ν*)). Mp = 55–57 °C. IR ν_max_ (KBr, cm^–1^): 2958, 2923, 2855, 1618, 1507, 1457, 1389, 1324, 1282, 1262, 1144, 1078, 1032, 881, 861, 847, 812, 786, 739, 618, 547. ^1^H NMR (300 MHz, CDCl_3_): *δ* 9.10 (s, 1H), 8.09 (d, *J* = 3.7, 1H), 7.01 (d, *J* = 3.7, 1H), 2.91 (d, *J* = 6.8, 2H), 1.78–1.68 (m, 1H), 1.48–1.29 (m, 8H), 0.98–0.89 (m, 6H). ^13^C NMR (100 MHz, CDCl_3_): *δ* 158.3, 151.1, 140.8, 136.7, 135.1, 131.8, 127.0, 126.5, 123.6, 111.1, 41.6, 34.4, 32.5, 28.9, 25.7, 23.0, 14.1, 10.9. HRMS (ESI-TOF), *m*/*z*: calcd for C_18_H_21_N_4_S_3_ [M + H]^+^, 389.0923; found, 389.0921. MS (EI, 70 eV), *m*/*z* (*I*, %): 390 ([M + 2]^+^, 3), 389 ([M + 1]^+^, 6), 388 ([M]^+^, 35), 360 (80), 332 (15), 261 (18), 248 (38), 233 (100), 69 (28), 57 (60), 41 (45), 29 (37).

4-Phenylbenzo[1,2-*d*:4,5-*d*′]bis([1,2,3]thiadiazole) (**8e**)

Yellow solid, 33 mg (68%, Procedure A), 30 mg (63%, Procedure B), eluent–CH_2_Cl_2_/hexane, 1:2 (*v*/*v*). R_f_ = 0.3 (CH_2_Cl_2_/hexane, 1:1, (*ν*/*ν*)). Mp = 203–205 °C. IR ν_max_ (KBr, cm^–1^): 1637, 1492, 1431, 1386, 1277, 1148, 1075, 893, 862, 813, 745, 696, 673, 623, 545, 523. ^1^H NMR (300 MHz, CDCl_3_): *δ* 9.28 (s, 1H), 7.99 (d, *J* = 6.7, 2H), 7.69–7.59 (m, 3H). ^13^C NMR (100 MHz, CDCl_3_): *δ* 157.9, 155.6, 140.8, 140.1, 136.9, 130.3, 129.9, 129.7, 129.3, 112.7. HRMS (ESI-TOF), *m*/*z*: calcd for C_12_H_7_N_4_S_2_ [M + H]^+^, 271.0107/ found, 271.0109. MS (EI, 70 eV), *m*/*z* (*I*, %): 270 ([M+]^+^, 3), 242 (58), 214 (26), 170 (23), 145 (90), 93 (20), 69 (100), 28 (40).

4-(*p*-Tolyl)benzo[1,2-*d*:4,5-*d*′]bis([1,2,3]thiadiazole) (**8f**)

Green solid, 32 mg (64%, Procedure A), 34 mg (68%, Procedure B), eluent–CH_2_Cl_2_/hexane, 1:2 (*v*/*v*). R_f_ = 0.3 (CH_2_Cl_2_/hexane, 1:1, (*ν*/*ν*)). Mp = 229–232 °C. IR ν_max_ (KBr, cm^–1^): 2925, 1639, 1609, 1507, 1427, 1379, 1331, 1317, 1291, 1275, 1192, 1147, 1120, 895, 865, 828, 804, 763, 716, 670, 609, 556, 536, 488. ^1^H NMR (300 MHz, CDCl_3_): *δ* 9.24 (s, 1H), 7.89 (d, *J* = 7.9, 2H), 7.46 (d, *J* = 7.9, 2H), 2.51 (s, 3H). ^13^C NMR (100 MHz, CDCl_3_): *δ* 157.8, 155.5, 140.6, 140.4, 139.8, 134.0, 130.0, 129.8, 129.4, 112.1, 21.3. HRMS (ESI-TOF), *m*/*z*: calcd for C_13_H_8_BrN_4_S_2_ [M + H]^+^, 285.0263; found, 285.0266. MS (EI, 70 eV), *m*/*z* (*I*, %): 284 ([M]^+^, 3), 256 (8), 227 (5), 159 (25), 139 (5), 93 (7), 69 (100), 63 (7), 51 (10), 39 (30), 28 (45), 18 (70).

4-(4-Methoxyphenyl)benzo[1,2-*d*:4,5-*d*′]bis([1,2,3]thiadiazole) (**8g**)

Orange solid, 37 mg (69%, Procedure A), 37 mg (70%, Procedure B), eluent–CH_2_Cl_2_/hexane, 1:2 (*v*/*v*). R_f_ = 0.2 (CH_2_Cl_2_/hexane, 1:1, (*ν*/*ν*)). Mp = 198–201 °C. IR ν_max_ (KBr, cm^–1^): 3076, 1609, 1509, 1457, 1430, 1383, 1300, 1279, 1262, 1178, 1150, 1116, 1030, 896, 863, 835, 806, 670, 540. ^1^H NMR (300 MHz, CDCl_3_): *δ* 9.22 (s, 1H), 7.97 (d, *J* = 8.8, 2H), 7.17 (d, *J* = 8.8, 2H), 3.95 (s, 3H). ^13^C NMR (100 MHz, CDCl_3_): *δ* 161.2, 158.0, 155.6, 140.8, 139.4, 131.2, 129.9, 129.2, 114.8, 112.0, 55.6. HRMS (ESI-TOF), *m*/*z*: calcd for C_13_H_9_N_4_OS_2_ [M + H]^+^, 301.0212; found, 301.0215. MS (EI, 70 eV), *m*/*z* (*I*, %): 302 ([M + 2]^+^, 3), 301 ([M + 1]^+^, 4), 300 ([M]^+^, 30), 272(50), 229 (45), 201 (25), 175 (80), 132 (65), 93 (35), 69 (100), 28 (30).

4-(benzo[1,2-*d*:4,5-*d*′]bis([1,2,3]thiadiazole)-4-yl)-*N*,*N*-diphenylaniline (**8h**)

Orange solid, 55 mg (70%, Procedure A), 50 mg (64%, Procedure B), eluent–CH_2_Cl_2_/hexane, 1:2 (*v*/*v*). R_f_ = 0.25 (CH_2_Cl_2_/hexane, 1:1, (*ν*/*ν*)). Mp = 213–215 °C. IR ν_max_ (KBr, cm^–1^): 1727, 1590, 1487, 1428, 1321, 1276, 1195, 1125, 1073, 894, 865, 835, 808, 748, 696, 624, 512. ^1^H NMR (300 MHz, CDCl_3_): *δ* 9.18 (s, 1H), 7.88 (d, *J* = 8.8, 2H), 7.35 (t, *J* = 7.8 Hz, 3H), 7.28–7.12 (m, 9H). ^13^C NMR (100 MHz, CDCl_3_): *δ* 158.0, 155.3, 150.0, 146.9, 140.8, 139.3, 130.6, 129.8, 129.6, 129.0, 125.7, 124.3, 121.4, 111.5. HRMS (ESI-TOF), *m*/*z*: calcd for C_24_H_15_N_5_S_2_ [M]^+^, 437.0763; found, 437.0757. MS (EI, 70 eV), *m*/*z* (*I*, %): 438 ([M + 1]^+^, 8), 437 ([M]^+^, 55), 409 (6), 381 (4), 312 (12), 168 (3), 69 (15), 18 (100).

#### 3.5.6. General Procedure for the Preparation of Arylated Benzo-bis-thiadiazoles **14** from 4-Bromobenzo[1,2-*d*:4,5-*d*′]bis([1,2,3]thiadiazole) **2** (Procedure D)

A mixture of 4-bromobenzo[1,2-*d*:4,5-*d*′]bis([1,2,3]thiadiazole) **2** (50 mg, 0.18 mmol), bromide or iodide **13a**–**d**,**f**–**j** (1.03 mmol), Pd(OAc)_2_ (17 mg, 0.076 mmol), (P(Bu^t^)_2_Me·HBF_4_) (41 mg, 0.18 mmol), pivalic acid (105 mg, 1.03 mmol), and K_2_CO_3_ (142 mg, 1.03 mmol) were added to an air-free flask, which was then purged in dry toluene (8 mL), degassed by argon, and heated at 110 °C in a sealed vial. On completion (monitored by TLC), the mixture was poured into water and extracted with CH_2_Cl_2_ (3 × 35 mL). The combined organic layers were washed with brine, dried over MgSO_4_, filtered, and concentrated under reduced pressure. The crude product was purified by column chromatography.

#### 3.5.7. General Procedure for the Preparation of Arylated Benzo-bis-thiadiazoles **14** under C-H Oxidative Coupling Conditions (Procedure E)

A mixture of 4-bromobenzo[1,2-*d*:4,5-*d*′]bis([1,2,3]thiadiazole) **2** (50 mg, 0.18 mmol), thiophene (0.36 mmol), Ag_2_O (83 mg, 0.36 mmol), and Pd(OAc)_2_ (6 mg, 0.027 mmol), were added to an air-free flask which, was then purged in dry DMSO (5 mL), degassed by argon, and heated at 90 °C in a sealed vial. On completion (monitored by TLC), the mixture was poured into water and extracted with CH_2_Cl_2_ (3 × 35 mL). The combined organic layers were washed with brine, dried over MgSO_4_, filtered, and concentrated under reduced pressure. The crude product was purified by column chromatography.

4-Bromo-8-(thiophen-2-yl)benzo[1,2-*d*:4,5-*d*′]bis([1,2,3]thiadiazole) (**14a**)

Yellow solid, 20 mg (32%, Procedure D), 19 mg (30%, Procedure E), eluent–CH_2_Cl_2_/hexane, 1:1 (*v*/*v*). R_f_ = 0.4 (CH_2_Cl_2_). Mp = 198–200 °C (lit. mp 198–200 °C [18]). The data of the ^1^H and ^13^C NMR spectra correspond to the literature data [18].

4-Bromo-8-(4-hexylthiophen-2-yl)benzo[1,2-*d*:4,5-*d*′]bis([1,2,3]thiadiazole) (**14b**)

Yellow solid, 31 mg (40%, Procedure D), 30 mg (38%, Procedure E), eluent–CH_2_Cl_2_/hexane, 1:2 (*v*/*v*). R_f_ = 0.6 (CH_2_Cl_2_/hexane, 1:1 (*v*/*v*)). Mp = 67–69 °C (lit. mp 67–69 °C [18]). The data of the ^1^H and ^13^C NMR spectra correspond to the literature data [18].

4-([2,20-Bithiophen]-5-yl)-8-bromobenzo[1,2-*d*:4,5-*d*′]bis([1,2,3]thiadiazole) (**14c**)

Red solid, 39 mg (50%, Procedure D), 37 mg (48%, Procedure E), eluent–CH_2_Cl_2_/hexane, 1:1 (*v*/*v*). R_f_ = 0.4 (CH_2_Cl_2_/hexane, 1:1 (*v*/*v*)). Mp = 130–132 °C (lit. mp 130–132 °C [18]). The data of the ^1^H and ^13^C NMR spectra correspond to the literature data [18].

4-Bromo-8-(5-(2-ethylhexyl)thiophen-2-yl)benzo[1,2-*d*:4,5-*d*′]bis([1,2,3]thiadiazole) (**14d**)

Yellow solid, 54 mg (65%, Procedure D), 57 mg (68%, Procedure E), eluent–CH_2_Cl_2_/hexane, 1:1 (*v*/*v*). R_f_ = 0.6 (CH_2_Cl_2_). Mp = 57–60 °C (lit. mp 57–60 °C [18]). The data of the ^1^H and ^13^C NMR spectra correspond to the literature data [18].

4-Bromo-8-phenylbenzo[1,2-*d*:4,5-*d*′]bis([1,2,3]thiadiazole) (**14e**)

Yellow solid, 32 mg (52%, procedure D), eluent–CH_2_Cl_2_/hexane, 1:1 (*v*/*v*). R_f_ = 0.4 (CH_2_Cl_2_). Mp = 163–165 °C (lit. mp 163–165 °C [18]). The data of the ^1^H and ^13^C NMR spectra correspond to the literature data [18].

4-Bromo-8-(p-tolyl)benzo[1,2-*d*:4,5-*d*′]bis([1,2,3]thiadiazole) (**14f**)

Yellow solid, 44 mg (68%, procedure D), eluent–CH_2_Cl_2_/hexane, 1:1 (*v*/*v*). R_f_ = 0.5 (CH_2_Cl_2_). Mp = 135–137 °C (lit. mp 135–137 °C [18]). The data of the ^1^H and ^13^C NMR spectra correspond to the literature data [18].

4-Bromo-8-(4-methoxyphenyl)benzo[1,2-*d*:4,5-*d*′]bis([1,2,3]thiadiazole) (**14g**)

Yellow solid, 41 mg (60%, procedure D), eluent–CH_2_Cl_2_/hexane, 1:1 (*v*/*v*). R_f_ = 0.2 (CH_2_Cl_2_). Mp = 125–127 °C (lit. mp 125–127 °C [18]). The data of the ^1^H and ^13^C NMR spectra correspond to the literature data [18].

4-(8-Bromobenzo[1,2-*d*:4,5-*d*′]bis([1,2,3]thiadiazole)-4-yl)-*N*,*N*-diphenylaniline (**14h**)

Red solid, 53 mg (58%, procedure D), eluent–CH_2_Cl_2_/hexane, 1:2 (*v*/*v*). R_f_ = 0.5 (CH_2_Cl_2_/hexane, 1:1 (*v*/*v*)). Mp = 165–168 °C. (lit. mp 165–168 °C [18]). The data of the ^1^H and ^13^C NMR spectra correspond to the literature data [18].

## 4. Conclusions

The aromaticity of benzo[1,2-*d*:4,5-*d*′]bis([1,2,3]thiadiazole) (**isoBBT**) and benzo[1,2-*c*:4,5-*c*′]bis[1,2,5]thiadiazole (**BBT**) heterocycles was confirmed by two modern criteria, EDDB and GIMIC, at the MP2 level of theory and additionally by UV-vis spectroscopy. The aromaticity degree of **isoBBT** derivatives is significantly weaker than that of **BBT** as evidenced by both π-EDDB**_H_** and IRCS values, and the delocalization in **BBT** is stronger in five-membered rings in contrast to **isoBBT**. The incorporation of bromine substituents into the **isoBBT** molecule enhances the electron-withdrawing properties that may increase its ability participate in nucleophilic aromatic substitution reactions. At the same time, the aromaticity of **isoBBT** derivatives practically does not change, and their reactivity to cross-coupling reactions is preserved, which is confirmed by the data of 4-bromobenzo[1,2-*d*:4,5-*d*′]bis([1,2,3]thiadiazole) (4-bromo**isoBBT**) reactions. The study of aromatic nucleophilic substitution of 4-bromo**isoBBT** showed that its reactivity is close to that of 4,8-dibromobenzo[1,2-*d*:4,5-*d*′]bis([1,2,3]thiadiazole; a series of 4-amino and 4-thio-substituted derivatives were obtained in high yields. A number of mono(het-)aryl-substituted **isoBBT** derivatives have been successfully synthesized from 4-bromo**isoBBT**. 4-Aryl**isoBBT** derivatives, not available by other methods, were prepared by several variants of palladium-catalyzed cross-coupling reactions, namely, Suzuki, Stille, and direct C–H arylation with 2-unsubstituted thiophenes. Selective methods have been developed for the synthesis of 4-bromo-8-aryl**isoBBT** derivatives by cross-coupling reactions of 4-bromo**isoBBT** with (het)aryl halides and an oxidative direct C–H arylation with thiophenes.

## Data Availability

Data are contained within the article or Appendix A.

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
