# Peer review of "Benzo[1,2-*d*:4,5-*d*′]bis([1,2,3]thiadiazole) and Its Bromo Derivatives: Molecular Structure and Reactivity"

_ijms, 2023, doi:10.3390/ijms24108835_

Round 1

Reviewer 1 Report

The article is devoted to the structure and reactivity of benzo-bis-thiadiazole derivatives. It looks like the research has been done carefully. However, before publishing in the International Journal of Molecular Sciences, the Authors should make some explanations for the following issues:

1. The description of the structure is very poor and needs revision, e.g. compare the presented structures with the structures of similar compounds available in the CSD database or literature.

2. The cifcheck report cif report shows the following alert ”PLAT112_ALERT_2_C ADDSYM Detects New (Pseudo) Symm. Elem A”. Did the Authors try to solve this problem whether (i.e. did the Platon software (addsym) was used?).

3. In cif file of compound 3 there is information that EADP commends were used. The use of EADP records should be documented in the experimental section of the manuscript.

4. The authors wrote, "…. the length of all chemical bonds falls in the range typical for thiadiazole derivatives". On what basis was this confirmed? There is no literature reference.

5. The Figure 3. - Instead of the letters a, b, c, give the numbers of the compounds.

6. The figure(s) with the 2D-fingerprint plots (HS) should be added.

7. Table 3.: What dictated the choice of these and no other solvents? Whether MeCN solvent was used in the substitution reaction of compound 2 with amines f and g? In the case of the substitution reaction of compound 2 with amine a, was an attempt made to use a solvent other than MeCN?

8. There are many typos and grammatical errors, e.g. line 85 -“Figire” it should be figure; no space between the word and the cited reference; “… the molecules of 1 and 3 occupies the positions..” it should be “occupy”, etc.

Often the verb is given in the wrong form, i.e. instead of the singular form, it is plural and vice versa.  Sometimes is a lack of articles: the and a/an.

Author Response

Response to Reviewer 1.

The authors are grateful to the reviewer for a kind and highly professional review.

Reviewer 1:

  1. The description of the structure is very poor and needs revision, e.g. compare the presented structures with the structures of similar compounds available in the CSD database or literature.

Authors:

There are a lack of compounds for the direct comparison. Nevertheless the most of thiadiazoles in Cambridge structural database [version of 2022] is characterized by similar structural parameters.

Reviewer 1:

  1. The cifcheck report cif report shows the following alert ”PLAT112_ALERT_2_C ADDSYM Detects New (Pseudo) Symm. Elem A”. Did the Authors try to solve this problem whether (i.e. did the Platon software (addsym) was used?).

Authors:

Yes, we have tried to use ADDSYMM. No any improvements was obtained. The output of checkcif is not mandatory. Despite of winning, in our current models for 1 and 3 we obtained low R-values and reasonable ADPs.

Reviewer 1:

  1. In cif file of compound 3 there is information that EADP commends were used. The use of EADP records should be documented in the experimental section of the manuscript.

Authors:

The reason is non-merohedral twinning. The corresponding explanation was added to Experimental section.

Reviewer 1:

  1. The authors wrote, "…. the length of all chemical bonds falls in the range typical for thiadiazole derivatives". On what basis was this confirmed? There is no literature reference.

Authors:

We use the “Tables of bond lengths determined by X-ray and neutron diffraction. Part 1. Bond lengths in organic compounds ” and Cambridge structural database [version of 2022] to reveal this fact. So we add these references to our list.

Reviewer 1:

  1. The Figure 3. - Instead of the letters a, b, c, give the numbers of the compounds.

Authors:

Corrected as suggested by the Reviewer.

Reviewer 1:

  1. The figure(s) with the 2D-fingerprint plots (HS) should be added.

Authors:

The figures of fingerprint plot were added to ESI.

Reviewer 1:

  1. Table 3.: What dictated the choice of these and no other solvents? Whether MeCN solvent was used in the substitution reaction of compound 2 with amines f and g? In the case of the substitution reaction of compound 2 with amine a, was an attempt made to use a solvent other than MeCN?

Authors:

The choice of MeCN as a solvent for nucleophilic aromatic substitution reactions with amines such as morpholine, piperidine, and pyrrolidine was due to the fact that when other solvents, such as methylene chloride or chloroform, were used, both at room temperature and at refluxing, no chemical reaction was observed. Aliphatic primary amines did not react with molecule 2 either in acetonitrile or in DMF: slow decomposition of the starting compound 2 into a mixture of unidentifiable compounds was observed.

Reviewer 1:

  1. There are many typos and grammatical errors, e.g. line 85 -“Figire” it should be figure; no space between the word and the cited reference; “… the molecules of 1 and 3 occupies the positions..” it should be “occupy”, etc

Authors:

Corrected as suggested by the Reviewer.

Reviewer 1:

Often the verb is given in the wrong form, i.e. instead of the singular form, it is plural and vice versa.  Sometimes is a lack of articles: the and a/an.

Authors:

Our manuscript was checked by a colleague fluent in English writing and the errors were corrected. Also we have carefully checked the manuscript and corrected the mistakes.

Reviewer 2 Report

This manuscript by Chmovzh et al. describes molecular structure and reactivity of benzo[1,2-d:4,5-d']bis([1,2,3]thiadiazole) and its bromo derivatives. This work is well planned, results and discussions sections are well written in this manuscript. Authors have well studied the structures of these compounds using X-ray diffraction analysis and ab-initio calculations. Authors nicely optimized various reaction conditions for the derivatization of these substrates in good to high yields. I would recommend this manuscript may be considered for publication in IJMS.

Author Response

Response to Reviewer 2.

The authors are grateful to the reviewer for highly professional review.

Reviewer 2:

Lines 72-80: In this block the authors talk about selectivity of arylation. It is not clear what kind of selectivity they mean in the case of mono-bromoisoBBT 2. The selectivity of aromatic nucleophilic substitution and cross-coupling reactions could be observed for non-brominated isoBBT 1 or dibromo-isoBBT 3.

Authors:

Our task was to find conditions for the selective substitution of hydrogen or bromine atoms in position 4 in order to obtain compounds containing a (het)aryl group in this position, and to use the remaining unsubstituted hydrogen or bromine atoms to obtain unsymmetrically substituted isoBBT derivatives, potentially interesting compounds for the synthesis of OLEDs and organic solar cells components.

Reviewer 2:

Table 3: "R1R2NH =" should be written instead of "R1R2N ="

Authors:

Corrected as suggested by the Reviewer.

Reviewer 2:

Table 3, entry 9: maybe DMF, not MeCN?

Authors:

Corrected as suggested by the Reviewer.

Reviewer 2:

Line 226: please add "of" between "reactivity" and "monobromide"

Authors:

Corrected as suggested by the Reviewer.

Reviewer 2:

Line 232: hexanethiol, not hexylthiol

Authors:

Corrected as suggested by the Reviewer.

Reviewer 2:

Line 269: A comparison of the Stille Suzuki reaction...

Authors:

Corrected as suggested by the Reviewer.

Reviewer 2:

Scheme 3: tris(isopropyl)silyl radical should be changed in compound 12.

Authors:

Corrected as suggested by the Reviewer.

Reviewer 2:

Line 343: "with" should be excluded

Authors:

Corrected as suggested by the Reviewer.

Reviewer 2:

Line 350: bicycle 2 change to tricycle 2

Authors:

Corrected as suggested by the Reviewer.

Reviewer 2:

Line 394: "in moderate yield", not "with a moderate yield"

Authors:

Corrected as suggested by the Reviewer.

Reviewer 2:

Line 413: add "of" after "instead"

Authors:

Corrected as suggested by the Reviewer.

Reviewer 2:

Line 420: change compound number to 2

Authors:

Corrected as suggested by the Reviewer.

Reviewer 2:

Line 437: "spectra were recorded on a Bruker AM-300 instrument (or apparatus)" sounds better.

Authors:

Corrected as suggested by the Reviewer.

Reviewer 2:

Lines 672 and 675: check compounds numbering

Authors:

Compounds numbers changed as suggested by the Reviewer.

Reviewer 2:

Line 743: "which was" should be deleted

Authors:

Corrected as suggested by the Reviewer.

Reviewer 2:

Minor editing of English language is required for sections 2.3-2.4.3. There are several sentences which are difficult to understand or should be revised: lines 280-283; 367-369 and so on. Please check at least these sections indicated above.

Authors:

Our manuscript was checked by a colleague fluent in English writing and the errors were corrected. Also we have carefully checked the manuscript and corrected the mistakes.

Reviewer 3 Report

In this manuscript Oleg Rakitin and coworkers studied molecular structure of isoBBT derivatives using X-ray analysis and ab initio calculations in order to reveal electron deficiency and aromaticity compared to previously studied isomeric BBT derivatives. They showed that introduction of the bromine atom into isoBBT molecule enhanced the electron-withdrawing properties, however did not affect reactivity. The authors have found that reactivities of 4-bromo- and 4,8-dibromo-isoBBT are quite close. A series of monoarylated isoBBT derivatives were synthesized by means of Pd-catalyzed cross-coupling reactions (Suzuki-Miyaura, Stille, direct oxidative CH-arylation). The syntheses are well-described and the structures of the products are confirmed by NMR and other spectral methods.

I think the manuscript deserves publication in IJMS, however some minor revision is needed. In addition, some editing of English language is required.

Lines 72-80: In this block the authors talk about selectivity of arylation. It is not clear what kind of selectivity they mean in the case of mono-bromoisoBBT 2. The selectivity of aromatic nucleophilic substitution and cross-coupling reactions could be observed for non-brominated isoBBT 1 or dibromo-isoBBT 3.

Table 3: "R1R2NH =" should be written instead of "R1R2N ="

Table 3, entry 9: maybe DMF, not MeCN?

Line 226: please add "of" between "reactivity" and "monobromide"

Line 232: hexanethiol, not hexylthiol

Line 269: A comparison of the Stille Suzuki reaction...

Scheme 3: tris(isopropyl)silyl radical should be changed in compound 12.

Line 343: "with" should be excluded

Line 350: bicycle 2 change to tricycle 2

Line 394: "in moderate yield", not "with a moderate yield"

Line 413: add "of" after "instead"

Line 420: change compound number to 2

Line 437: "spectra were recorded on a Bruker AM-300 instrument (or apparatus)" sounds better

Lines 672 and 675: check compounds numbering

Line 743: "which was" should be deleted

Minor editing of English language is required for sections 2.3-2.4.3. There are several sentences which are difficult to understand or should be revised: lines 280-283; 367-369 and so on. Please check at least these sections indicated above.

Author Response

Response to Reviewer 3.

The authors are grateful to the reviewer for highly professional review.

Reviewer 3:

Query#1

The abstract was found to be too pale, I find it very short and not incisive and considering also that the abstract is the calling card of the article, I suggest to the authors to improve and specify the main purpose of the article, because in the present form is not sufficient informative about the aims that the author want to present in the article.

Authors:

The abstract has been rewritten in accordance with the suggestions of the reviewer.

Reviewer 3:

Query#2

The introduction was found to be too confused, for instance, the authors have introduced the benzothiadiazoles and their derivatives, for their excellent properties, such as their strong electron-withdrawing properties. Indeed, sulfur-containing heterocycles are often involved in attractive nonbonding interactions that play an important role in the control of molecular conformation. The presence of the low-lying C−S σ* or C-N σ* orbitals. The small regions of low electron density present on the sulfur/nitrogen atom, known as σ-holes, are often involved in drug–target interactions, thus improving the affinity toward the biological. In my opinion the authors should also report the different application of nitrogen electron-withdrawing system pharmaceutical fields, at this purpose I suggest to the authors to cite this update articles:

-Li Petri, G., Pecoraro, C., Randazzo, O., Zoppi, S., Cascioferro, S. M., Parrino, B., Carbone, D., El Hassouni, B., Cavazzoni, A., Zaffaroni, N., Cirrincione, G., Diana, P., Peters, G. J., & Giovannetti, E. (2020). New Imidazo[2,1-b][1,3,4]Thiadiazole Derivatives Inhibit FAK Phosphorylation and Potentiate the Antiproliferative Effects of Gemcitabine Through Modulation of the Human Equilibrative Nucleoside Transporter-1 in Peritoneal Mesothelioma. Anticancer research40(9), 4913–4919. https://doi.org/10.21873/anticanres.14494

- Beno BR, Yeung KS, Bartberger MD, Pennington LD, Meanwell NA. A Survey of the Role of Noncovalent Sulfur Interactions in Drug Design. J Med Chem. 2015;58(11):4383-4438. doi:10.1021/jm501853m

Authors:

The Introduction has been rewritten in accordance with the suggestions of the reviewer.

Reviewer 3:

Comments on the Quality of English Language

A careful revision of the English form and language is recommended for the publication.

Authors:

Our manuscript was checked by a colleague fluent in English writing and the errors were corrected. Also we have carefully checked the manuscript and corrected the mistakes.

Reviewer 4 Report

The manuscript entitled: “Benzo[1,2-d:4,5-d']bis([1,2,3]thiadiazole) and its bromo derivatives: molecular structure and reactivity”, is a scientific article, which discusses the role of fused heterocyclic systems containing nitrogen atoms endowed by acceptors properties.

This article could improve to the existing scientific literature about the electron properties and reactivity of fused heterocycles.

However, the present scientific article presents several issues, I would like to suggest to the authors to make some revisions and improvements for the publication.

Recommendation for revision:

Query#1

The abstract was found to be too pale, I find it very short and not incisive and considering also that the abstract is the calling card of the article, I suggest to the authors to improve and specify the main purpose of the article, because in the present form is not sufficient informative about the aims that the author want to present in the article.

Query#2

The introduction was found to be too confused, for instance, the authors have introduced the benzothiadiazoles and their derivatives, for their excellent properties, such as their strong electron-withdrawing properties. Indeed, sulfur-containing heterocycles are often involved in attractive nonbonding interactions that play an important role in the control of molecular conformation. The presence of the low-lying CS σ* or C-N σ* orbitals. The small regions of low electron density present on the sulfur/nitrogen atom, known as σ-holes, are often involved in drugtarget interactions, thus improving the affinity toward the biological. In my opinion the authors should also report the different application of nitrogen electron-withdrawing system pharmaceutical fields, at this purpose I suggest to the authors to cite this update articles:

-Li Petri, G., Pecoraro, C., Randazzo, O., Zoppi, S., Cascioferro, S. M., Parrino, B., Carbone, D., El Hassouni, B., Cavazzoni, A., Zaffaroni, N., Cirrincione, G., Diana, P., Peters, G. J., & Giovannetti, E. (2020). New Imidazo[2,1-b][1,3,4]Thiadiazole Derivatives Inhibit FAK Phosphorylation and Potentiate the Antiproliferative Effects of Gemcitabine Through Modulation of the Human Equilibrative Nucleoside Transporter-1 in Peritoneal Mesothelioma. Anticancer research40(9), 4913–4919. https://doi.org/10.21873/anticanres.14494

- Beno BR, Yeung KS, Bartberger MD, Pennington LD, Meanwell NA. A Survey of the Role of Noncovalent Sulfur Interactions in Drug Design. J Med Chem. 2015;58(11):4383-4438. doi:10.1021/jm501853m

A careful revision of the English form and language is recommended for the publication.

Author Response

Response to Reviewer 4.

The authors are grateful to the reviewer for highly professional review.

Reviewer 4:

  1. I think it's worth adding a few references in the first paragraph to reinforce the importance of the study of aromaticity in organic chemistry.

«Aromaticity and reactivity of heterocyclic compounds are one of the most studied problems in organic chemistry. The properties of monocyclic heterocycles have been well studied for a long time, while fused heterocyclic systems often face a number of problems with their aromaticity/antiaromaticity and the consequently different reactivity. Fused heterocyclic systems containing a large number of nitrogen and chalcogen (mainly sulfur) atoms in cycles, which have pronounced acceptor properties, have attracted particular interest in recent years.» These phrases should be backed up with links.

Authors:

Necessary references have been added to the Introduction in accordance with the suggestions of the reviewer.

Reviewer 4:

  1. Schemes 3,4,5,6,7.

Add letters to compounds:

5 change on 5a-g; 7 change on 7a-h; 9 change on 9a-h; 13/13a-h; 15/15a-d

  1. In schemes 4,5,6 – “h” should be in bold

Authors:

Corrected as suggested by the Reviewer.

Response to Language Editor.

Editor:

If the reviewers or editor recommended English language editing, this can be arranged by MDPI. Note that language editing by MDPI is not compulsory, nor does it guarantee that your manuscript will eventually be accepted for publication. Click on the link for more information and to request a quotation.

Authors:

Our manuscript was checked by a colleague fluent in English writing and the errors were corrected. Also we have carefully checked the manuscript and corrected the mistakes.

Reviewer 5 Report

In this manuscript authors reported thier study of the molecular structure of benzo[1,2-d:4,5-d']bis([1,2,3]thiadiazole and its bromo derivative. The authors revealed that the introduction of brome into the molecule of the initial bis([1,2, 3]thiadiazole increases its electrophilicity while saving aromaticity, which made it possible to synthesize a number of 4-R derivatives using nucleophilic substitution and cross-coupling reactions.

These results provide useful information for readers. Since this study was adequately performed, I think this manuscript is acceptable for publication in IJMS. However, several issues should be addressed before it.

1.      I think it's worth adding a few references in the first paragraph to reinforce the importance of the study of aromaticity in organic chemistry.

«Aromaticity and reactivity of heterocyclic compounds are one of the most studied problems in organic chemistry. The properties of monocyclic heterocycles have been well studied for a long time, while fused heterocyclic systems often face a number of problems with their aromaticity/antiaromaticity and the consequently different reactivity. Fused  heterocyclic systems containing a large number of nitrogen and chalcogen (mainly sulfur) atoms in cycles, which have pronounced acceptor properties, have attracted particular  interest in recent years.»

These phrases should be backed up with links.

2.      Schemes 3,4,5,6,7.

Add letters to compounds:

5 change on 5a-g; 7 change on 7a-h; 9 change on 9a-h; 13/13a-h; 15/15a-d

3.      In schemes 4,5,6 –  “h” should be in bold

Author Response

(The authors gave the same response as above.)

Round 2

Reviewer 1 Report

Most of my comments have been taken into account. However, in the experimental part, I did not find information that EADP instructions were used. The authors have added figures with the 2D-fingerprint plots (HS), but there are no references and commentary for them in the text There are still some typos and grammatical errors. 

There are still a few grammatical errors. 

Author Response

Response to Reviewer 1.

Reviewer 1:

Most of my comments have been taken into account. However, in the experimental part, I did not find information that EADP instructions were used. The authors have added figures with the 2D-fingerprint plots (HS), but there are no references and commentary for them in the text There are still some typos and grammatical errors.

Authors:

The cif file was re-refined without EADP instruction. As result, we re-deposited CIF file for CCDC and generated new checkcif report. In addition, comments about 2D fingerplots were added to the main text (page 3).

We corrected typos and grammatical errors in the manuscript.